# Comparing costs and climate impacts of various electric vehicle charging systems across the United States

Noah Horesh [1], David A. Trinko[1] & Jason C. Quinn [1] ✉

The seamless adoption of electric vehicles (EVs) in the United States necessitates the development of extensive and effective charging infrastructure. Various charging systems have been proposed, including Direct Current Fast Charging, Battery Swapping, and Dynamic Wireless Power Transfer. While many studies have evaluated the charging costs and greenhouse gas (GHG) intensity of EVs, a comprehensive analysis comparing these systems and their implications across vehicle categories remains unexplored. This study compares the total cost of ownership (TCO) and GHG-intensity of EVs using these charging systems. Based on nationwide infrastructure deployment simulations, the change to TCO from adopting EVs varies by scenario, vehicle category, and location, with local fuel prices, electricity prices, and traffic volumes dramatically impacting results. Further, EV GHG-intensity depends on local electricity mixes and infrastructure utilizations. This research highlights the responsiveness of EV benefits resulting from technology advancements, deployment decisions, and policymaking.

The United States (U.S.) is currently undertaking an ambitious initiative to deploy public charging infrastructure to facilitate the widespread adoption of electric vehicles (EVs) necessary for achieving climate targets[1]. As EVs continue to gain popularity in all vehicle classes, ensuring uninterrupted transportation has become a critical objective for policymakers and stakeholders[1,2]. While initial efforts have focused on deploying Level 2 and Direct Current Fast Charging (DCFC) infrastructure[3], a significant challenge lies in the charging time required to replenish EV batteries[2]. Long charging times pose potential inconveniences for EV drivers, particularly those embarking on long journeys or requiring urgent charging[2]. Addressing this issue necessitates the implementation of charging systems capable of fulfilling consumer needs[2,4]. The electrification of medium-duty vehicles (MDVs) and heavy-duty vehicles (HDVs)[4] changes the infrastructure needs to accommodate a broader range of vehicles beyond passenger cars and light-duty trucks (LDTs)[5]. Consequently, it is imperative to transition towards technologies that enable short dwell times for all vehicle types, such as 350-kilowatt (kW) DCFC, Battery Swapping (BSS), or Dynamic Wireless Power Transfer (DWPT)[5].

Each of these technologies presents a distinct set of benefits and challenges. DCFC follows a similar model to traditional liquid refueling and allows for scalability by increasing the number of stations according to demand. However, the intermittent high-power loads of DCFC present challenges to the electrical grid and costs to the consumer[6]. In contrast, BSS optimizes grid loads by charging batteries before they are swapped, but their successful implementation relies on battery standardization and addressing social challenges related to battery ownership[5,7]. In addition, BSS requires two different sizes: a small size for cars and LDTs and a large size for MDVs and HDVs[8]. Despite these challenges, BSS offers reduced dwell times compared to DCFC, making their dwell time comparable to that of internal combustion engine (ICEV) and hydrogen fuel cell vehicles[9,10]. Alternatively, DWPT inductively charges vehicles while they are in motion using embedded electronics in the roadway, effectively enabling smaller batteries and eliminating the need for vehicles to stop between destinations for recharging[11]. DWPT, however, may cause traffic disruptions during roadway replacements, has limited deployment history, and is capital intensive[12,13]. Despite the well-understood performance of these technologies, there remains a meaningful gap in

[1]Department of Mechanical Engineering, Colorado State University, 1374 Campus Delivery, Fort Collins, CO, USA. ✉e-mail: Jason.Quinn@colostate.edu

understanding the economic and environmental implications that would arise from their widescale deployment.

Current research on the economic and environmental impacts of EV charging systems is often narrowly focused, circumscribed by location, implementation scale, analysis scope, and scenario range. For instance, Wood et al. (2017) concentrated on the deployment of charging stations for varying levels of EV adoption without considering costs or greenhouse gas (GHG) emissions[14]. Muratori et al. (2019) examined variations in electricity costs for DCFC, finding utilization rates and locations meaningfully affect electricity costs but did not account for total charging costs with the exclusion of capital and operational expenses[15]. Mulrow and Grubert (2023) investigated the GHG emissions linked with traditional EV infrastructure, including DCFC, finding the embodied emissions of such infrastructure to be minimal[16].

Conversely, the environmental impact of infrastructure emissions from DWPT and BSS remains unclear. Marmiroli et al. (2019) evaluated the GHG-intensity of DWPT infrastructure per distance, omitting an evaluation based on energy consumption[17]. Similarly, Limb et al. (2018) simulated the deployment of DWPT, analyzing charging costs and electricity emissions without considering the total cost of ownership (TCO) or lifecycle emissions[11]. While BSS has been identified as potentially economically viable[18], its broader economic and environmental assessments are lacking. In summary, although there exists a body of literature that partially addresses the costs and GHG emissions associated with EV charging systems, there has yet to be a comprehensive study that evaluates these aspects holistically, compares the three charging technologies of BSS, DCFC, and DWPT, or considers their implications across different vehicle categories.

This study addresses this gap by simulating the nationwide deployment of DCFC, BSS, and DWPT and assessing the GHG-intensity and TCO of EVs utilizing these systems. By leveraging geospatially resolved charging demand, emissions, and cost data, this study determines location-specific sustainability outcomes. Deployment scenarios for the charging systems, spanning from 2031 to 2050, are formulated based on the geospatial demand derived from traffic data forecasts[19] and three EV adoption scenarios (Supplementary Figs. 1 and 2): optimistic, baseline, and conservative[4,20]. DCFC and BSS charging infrastructure is placed at existing DCFC sites, gas stations, and surface parking lots near grid interconnections. DCFC and BSS charging systems are modeled to represent a relatively small portion of EV charging, as the primary reliance remains on home or fleet charging. In contrast, DWPT infrastructure is deployed along major roadways to ensure that EVs can maintain their state of charge, consequently requiring a fixed amount of infrastructure per kilometer of roadway and providing a large portion of EV charging (see "Methods")[12]. Specifically, in optimistic and baseline EV adoption scenarios, DWPT is deployed on interstates, freeways, and principal arterial roads, thereby reducing the required EV battery sizes to a range of 56 km (35 miles) (Supplementary Fig. 3)[11]. However, with fewer EVs, the conservative EV adoption scenario assumes that only interstates are electrified, necessitating the use of full-size batteries for EVs. Simulation results are used to determine the levelized cost of charging and GHG-intensity for each DCFC site, BSS site, and DWPT roadway, accounting for each location's design, utilization, electricity costs, and electricity mix. The levelized cost of charging is used to determine the TCO for EVs, which is then compared to ICEVs and hybrid electric vehicles (HEVs) over a 10-year period per vehicle-kilometer traveled (VKT)[21]. Finally, to account for the uncertainty of evolving variables that highly influence TCO and GHG-intensity, optimistic, baseline, and conservative scenarios are modeled for electricity mixes, capital costs, electricity prices, and fuel prices (Supplementary Figs. 4 and 5). These scenarios result in a total of 81 TCO and 9 GHG-intensity comparisons for each charging system. The findings reveal meaningful variability in TCO and GHG-intensity benefits of vehicle

electrification, with impacts heavily dependent on the chosen charging system, location, vehicle category, and scenario, emphasizing the need for careful technology selection and innovation.

## Results
### Cost savings from EV adoption
The change in combined TCO when switching from ICEVs to EVs is illustrated on a county level in Fig. 1, which displays the aggregated results specific to the chosen scenarios for EV adoption, capital costs, electricity prices, and fuel prices. The findings suggest that the economic impacts of vehicle electrification in the U.S. are location-dependent and subject to variable changes, emphasizing the necessity of a dynamic visual. The change in TCO as a result of EV adoption from 2031 to 2050 is presented for baseline scenarios as a percentage in Fig. 1a–c and in billions (B) of 2022 U.S. Dollars (USD) in Fig. 1d–f.

Depending on the assumptions for each technology, the TCO of EVs can vary from being more favorable than ICEVs to less favorable, as demonstrated in Fig. 1 when combining conservative fuel prices with optimistic EV adoption, capital costs, and electricity prices, or vice versa. Moreover, the change to the TCO by switching from ICEVs to EVs varies depending on the location, with local fuel prices, electricity prices, and traffic volumes playing a considerable role in the change. In highly trafficked areas, substantial reductions in TCO are typically observed both as a percentage and in USD. Conversely, low-traffic areas typically show a relative increase in TCO, although the national impact in terms of USD remains limited due to fewer VKT. For instance, DWPT demonstrates the largest range of TCO change (−31% to +429%) and charging costs (Supplementary Fig. 8), primarily due to the heavy dependence on infrastructure utilization for upfront capital cost recovery. Thus, the capital cost allocation for using DWPT roadways may need to be based on a national average to ensure price equity. The charging cost for DCFC is heavily influenced by local peak demand prices (USD/kW per month), allocated based on site utilization (Supplementary Fig. 9). In contrast, BSS exhibits minimal price variability within a state by optimizing charging times to minimize electricity costs (Supplementary Fig. 9).

Although the charging cost for BSS has limited geographical variability, it is highly dependent on the assumptions around capital cost and EV adoption. Specifically, BSS has large capital costs, which can be reduced on a per kilowatt-hour (kWh) dispensed basis through lower battery prices (90–150 USD/kWh[22]) and greater utilization. Moreover, DWPT exhibits a wide range of capital costs (0.94–5.4 million USD/lane-kilometer[11,12,23]) across different scenarios due to the early-development status of the technology. Consequently, DWPT can have the highest or the lowest capital cost per kWh dispensed (Supplementary Fig. 9). This highlights the opportunity for cost reduction in DWPT, driven by the potential to decrease infrastructure capital expenses and its high utilization capacity. In contrast, DCFC incurs the highest electricity costs, mainly due to demand charges that are challenging to reduce given the requirement of supplying electricity at high power levels simultaneously with urgent charging. Further, projected electricity price scenarios exhibit minimal variations (Supplementary Fig. 4). Conversely, the energy or fuel price scenarios for ICEVs are highly variable (Supplementary Fig. 5) and impact whether EVs are economically favorable. The EV adoption scenarios, however, pose ambiguity as charging costs typically decrease with greater EV adoption, yet the combined change to the TCO from all vehicle categories can increase due to a higher portion of EVs in a vehicle category where EVs are more expensive than ICEVs.

The TCO of EVs can be lower than that of ICEVs for cars and LDTs but are in general higher for MDVs and HDVs, as illustrated in Fig. 2 with results reactive to the selected scenarios. Figure 2 presents the aggregated TCO at the national level for ICEVs, HEVs, DCFC-EVs, BSS-EVs, and DWPT-EVs under baseline scenarios. The variation of TCO

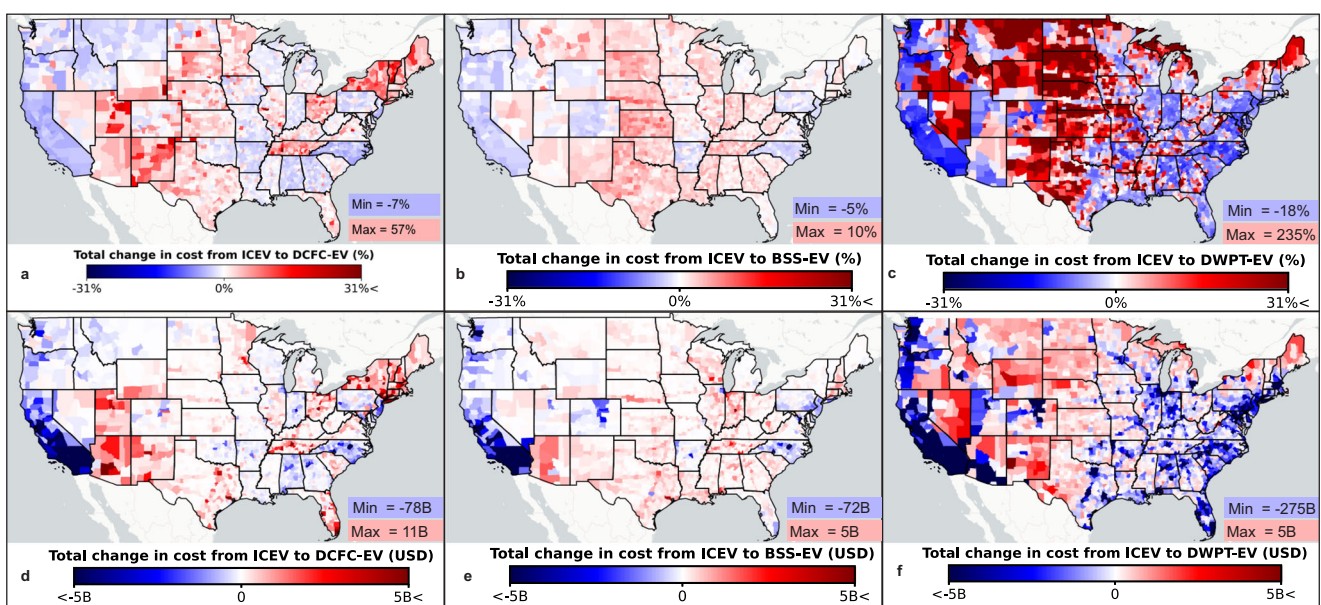

**Fig. 1 | Total change in cost due to electric vehicle (EV) adoption.** County level results are presented for the change in total cost of ownership due to the transition from internal combustion engine vehicles (ICEVs) to EVs (**a**–**c**) as a percentage and (**d**–**f**) in billions (B) of 2022 United States Dollars (USD). Each map corresponds to EVs charged via (**a, d**) Direct Current Fast Charging (DCFC), (**b, e**) Battery Swapping (BSS), and (**c, f**) Dynamic Wireless Power Transfer (DWPT). The baseline scenarios are shown in this figure with all scenarios shown in the repository. Source data are provided as a Source data file and base map layer is available from OpenStreetMap (openstreetmap.org/copyright).

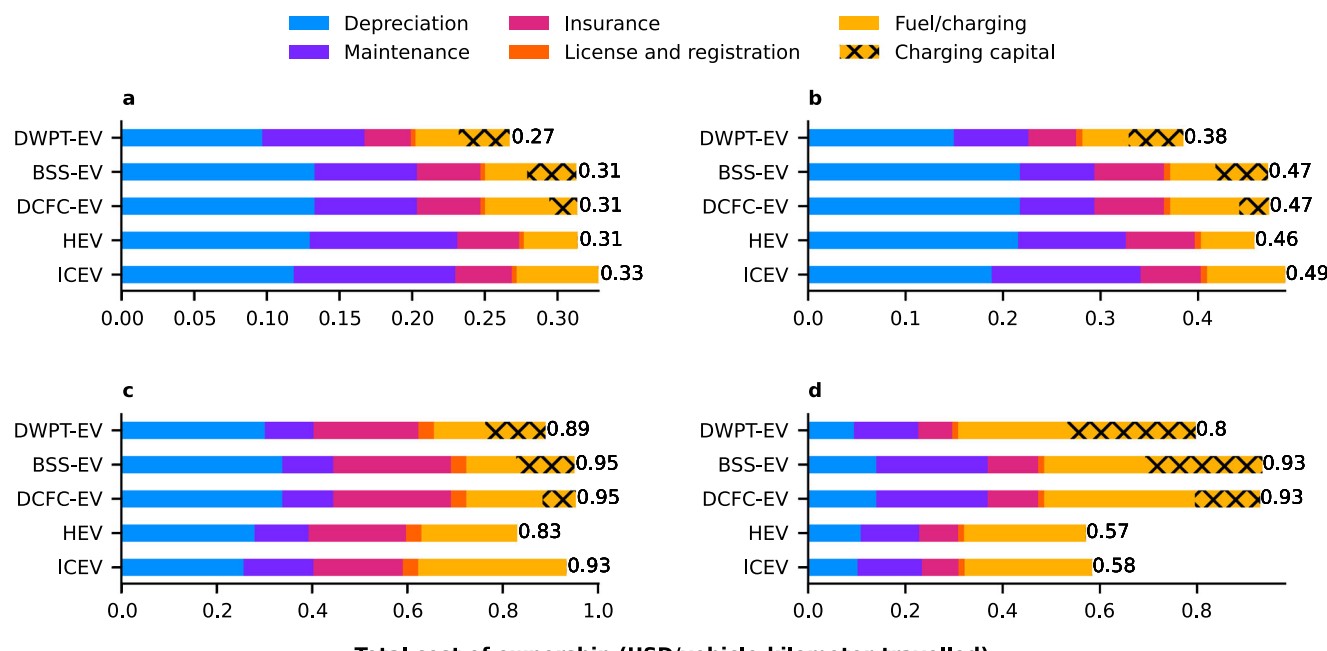

**Fig. 2 | Breakdown of the 10-year total cost of ownership.** Results are presented for an average **a** passenger car, **b** light-duty truck, **c** medium-duty vehicle, and **d** heavy-duty vehicle in the contiguous United States. The vehicle types include electric vehicles (EVs) charged via Direct Current Fast Charging (DCFC-EV), Battery Swapping (BSS-EV), and Dynamic Wireless Power Transfer (DWPT-EV). The EVs are compared to an average internal combustion engine vehicle (ICEV) and hybrid electric vehicle (HEV) from each vehicle category. 2022 United States Dollars (USD). The baseline scenarios are shown in this figure with all scenarios shown in the repository. Source data are provided as a Source data file.

among vehicle categories is due to the distinct contributions of each cost component to the overall TCO of the vehicles.

The breakdown of the TCO in Fig. 2 reveals several noteworthy findings. Depreciation emerges as a major cost contributor for cars, LDTs, and MDVs across all technologies. When comparing HEVs to ICEVs, there is a trade-off between higher depreciation costs and reduced maintenance and fuel costs for HEVs. Similarly, there are lower maintenance costs and higher depreciation costs for EVs in the car, LDT, and MDV categories. In contrast, EV maintenance costs are more expensive for HDVs due to expensive battery replacements within the initial 10-year lifespan, resulting from the combined factors of higher annual VKT and limited battery cycle life (see "Methods"). As

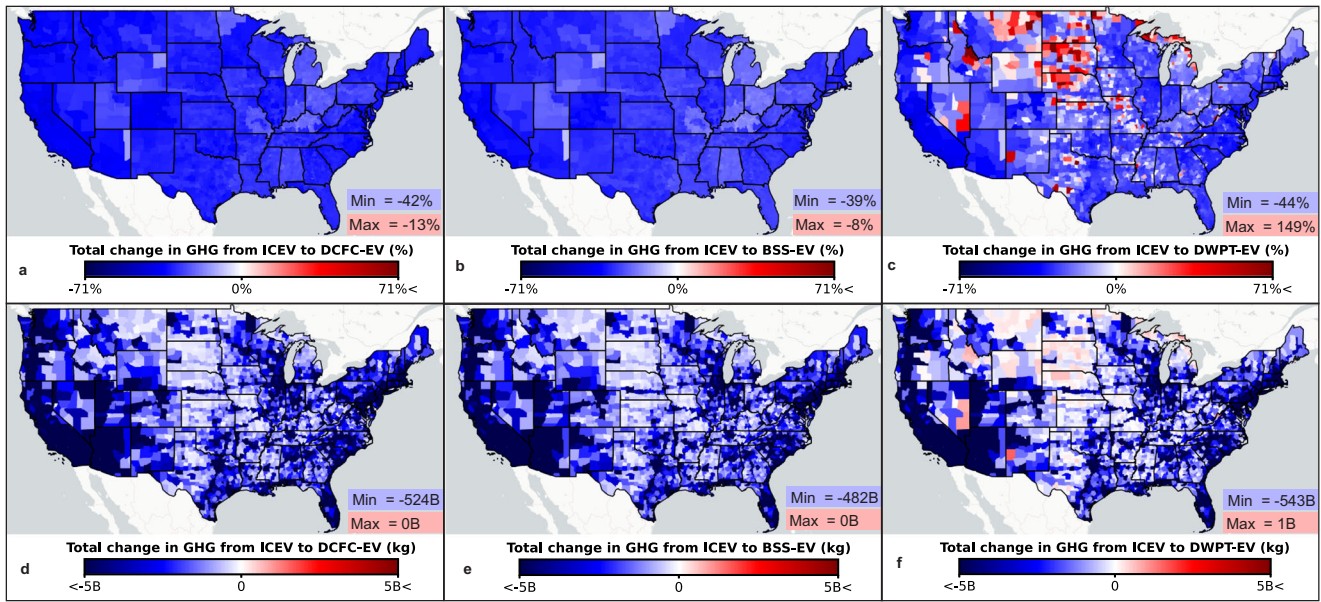

**Fig. 3 | Total change to greenhouse gas (GHG) emissions from electric vehicle (EV) adoption.** The maps are for the change in GHG emissions of on-road vehicle transportation in United States counties by switching from internal combustion engine vehicles (ICEVs) to EVs charged via (**a, d**) Direct Current Fast Charging (DCFC), (**b, e**) Battery Swapping (BSS), and (**c, f**) Dynamic Wireless Power Transfer (DWPT). The results are presented as (**a–c**) a percentage and (**d–f**) in billions (B) of kilograms (kg) of carbon dioxide equivalent. The baseline scenarios are shown in this figure with all scenarios shown in the repository. Source data are provided as a Source data file and base map layer is available from OpenStreetMap (open-streetmap.org/copyright).

a result, the DWPT-EV, with a reduced battery size that effectively lowers depreciation and maintenance costs, is the only electric HDV to demonstrate a cost advantage over an ICEV and HEV. Moreover, electric MDVs charged through DCFC or BSS exhibit cost advantages over ICEVs and HEVs solely in scenarios with high fuel prices, whereas MDVs charged via DWPT can achieve lower costs across all fuel price scenarios.

### Reduction to greenhouse gas emissions from EV adoption

The change in GHG emissions resulting from EV adoption is examined at the county level in Fig. 3, and the results reflect the optimistic, baseline, or conservative scenarios selected for EV adoption and electricity mixes. The findings highlight the influence of the grid mix and infrastructure utilization on GHG emissions changes from 2031 to 2050, presented both as percentages (Fig. 3a–c) and in kilograms (kg) of carbon dioxide equivalent ($CO_{2e}$) (Fig. 3d–f) for the baseline scenarios.

The percent change in GHG emissions is predominantly influenced by the local electricity mix for DCFC-EVs and BSS-EVs, while for DWPT-EVs, it is dependent on infrastructure utilization, as depicted in Fig. 3. In numerous locations, the scenarios for electricity mix and EV adoption change whether EVs increase or reduce transportation GHG emissions. Deploying DWPT in areas with lower utilization may increase the county GHG emissions by up to 167% but these areas have a limited national impact in terms of kg of $CO_{2e}$ due to fewer VKT. Similarly, certain areas may experience an increase in vehicle GHG emissions due to an electricity mix with a high carbon intensity, which depends on the scenario. Conversely, when infrastructure utilization is high and charging emissions are small due to a clean grid, the reduction in county-level GHG emissions can reach up to 71%.

Overall, EVs have a lower GHG-intensity nationally than ICEVs and HEVs across all scenarios and vehicle categories, as demonstrated in Fig. 4 for baseline scenarios. The breakdown of GHG-intensity reveals that the contribution of time-of-day (Supplementary Figs. 6 and 7) charging emissions is similar for all EV charging systems, while vehicle and infrastructure emissions differ.

The breakdown presented in Fig. 4 reveals that infrastructure emissions have a minimal contribution for DCFC-EVs, whereas they become substantial for BSS-EVs and DWPT-EVs, primarily due to the embodied emissions associated with high battery and concrete usage, respectively. Notably, the vehicle emissions reduction achieved through a reduced battery size in DWPT-EVs does not fully offset the infrastructure emissions when compared to DCFC-EVs. In addition, HEVs exhibit considerable emissions reductions compared to ICEVs, although their emissions are still much higher than those of EVs, even in scenarios with conservative EV adoption and electricity mix. This highlights the critical importance of zero emissions vehicles in decarbonizing the transportation sector. The magnitude of this impact depends on both the decarbonization of the electricity mix and the level of EV adoption, as illustrated in Fig. 4.

### Discussion

The transition from ICEVs to EVs could lead to remarkable changes to transportation TCO and GHG emissions, electricity grid infrastructure, and automotive manufacturing. The results of this study show that from 2031 to 2050, compared to ICEVs, on-road transportation costs can change by −22% to +11% and GHG emissions can change by −53% to −19% depending on various EV adoption and technology scenarios. Compared to 2022's TCO (Supplementary Fig. 10)[21], EV scenarios for 2031 to 2050 exhibit notably lower depreciation costs due to cheaper battery technologies but encounter higher charging costs from the premium associated with public charging infrastructure. Taking into account an overall increase in VKT[24], which remains consistent across scenarios without considering changes in user behavior due to cost or technology shifts, the reduction in U.S. GHG emissions by 2050, relative to the baseline year of 2022, is estimated to span from 1% to 58% depending on the scenario (Supplementary Fig. 11). This highlights the intricate challenges associated with emission reduction efforts amidst potential rises in vehicle usage.

During this period, the annual average electricity generation would increase by 16% to 38% relative to 2022 levels[25], potentially requiring utilities to upgrade grid infrastructure capacities. Controlling the charging loads from BSS and DWPT could reduce the need for

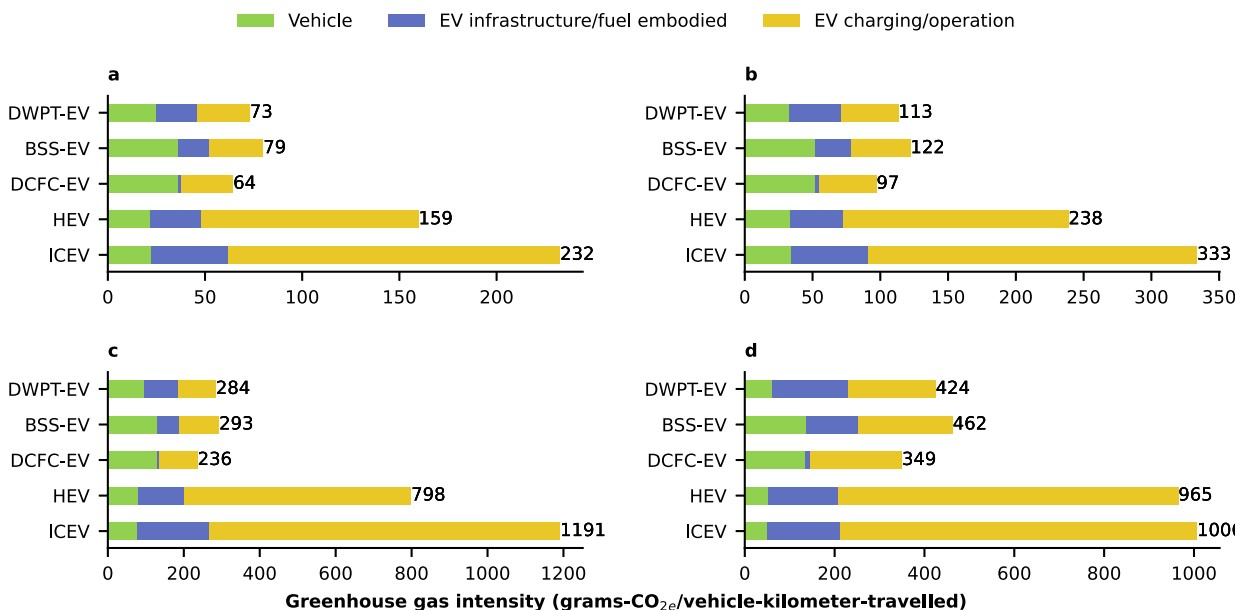

**Fig. 4 | Breakdown of the lifetime greenhouse gas intensity.** Results are for an average **a** passenger car, **b** light-duty truck, **c** medium-duty vehicle, and **d** heavy-duty vehicle in the contiguous United States. The vehicle scenarios include electric vehicles (EVs) charged via Direct Current Fast Charging (DCFC-EV), Battery Swapping (BSS-EV), and Dynamic Wireless Power Transfer (DWPT-EV). Results are compared to an internal combustion engine vehicle (ICEV) and hybrid electric vehicle (HEV) from each vehicle category. Carbon dioxide equivalent ($CO_{2e}$). The baseline scenarios are shown in this figure with all scenarios shown in the repository. Source data are provided as a Source data file.

grid capacity upgrades, in contrast to DCFC[8,26]. Furthermore, the total (2031–2050) anticipated battery production needed for full-size EV batteries ranges from 13 to 31 terawatt-hours, highlighting the necessity for major expansions to production capacity compared to the global production of 700 gigawatt-hours in 2022[27]. However, by using a smaller battery size, DWPT could reduce the battery production needed by 79%. This reduction could alleviate the global resource and manufacturing constraints associated with battery production[28].

DWPT, however, requires considerably higher capital investment at 134B to 1.7 trillion USD, as compared to 41B to 115B USD for BSS and 23B to 52B USD for DCFC. Notably, DWPT provides a much larger share of charging for EVs at 35–61% for cars, 32–61% for LDTs, 29–56% for MDVs, and 67–83% for HDVs. In comparison, DCFC and BSS are estimated to supply 5% of charging for cars and LDTs, 0.6% for MDVs, and 14% for HDVs. Therefore, DCFC and BSS deployment assumes EVs will primarily use home, workplace, or fleet charging[29,30].

DCFC technology emerges as the most advanced in terms of readiness. DCFC's early adoption within the market was facilitated through financing from charging providers, automakers, and federal funds. Presently, DCFC maintains a substantial presence in the car and LDT sectors. However, less than 1% of DCFC charging stations are accessible to MDVs and HDVs[3], presenting an opportunity for market disruption by BSS or DWPT.

Despite this potential, the widespread deployment of BSS and DWPT faces substantial challenges. BSS has seen limited deployment within the U.S. market but boasts over two thousand (k) stations in China, predominantly installed by the automaker NIO[31]. This demonstrates the technological maturity of BSS, though with minimal market penetration. Further, the deployment of BSS in China for MDVs and HDVs showcases its adaptability to various vehicle requirements[32]. The early adoption of BSS in China has underscored the necessity for battery standardization—regarding size, shape, and attachment mechanisms—to facilitate efficient capital recovery and ensure vehicle compatibility[5]. Nonetheless, the reluctance of U.S. automakers to forego proprietary battery technologies and vehicle owners' resistance to battery sharing present significant social obstacles[7]. Consequently, the widespread acceptance of BSS in the U.S. appears improbable between 2030 and 2050 for cars and LDTs, with social barriers rather than technological readiness posing the primary hindrance.

Conversely, DWPT faces challenges related to its untested technological readiness and uncertain consumer receptivity. Electreon initiated the first on-road DWPT pilot in the U.S. in 2023, in Michigan, with further pilots by Magment GmbH and ENRX scheduled for 2024 in Indiana and Florida, respectively. Thus, the limited data from these deployments does not yet substantiate the reliability and general applicability of DWPT for widespread use. In addition, the lack of standardized protocols for DWPT impedes its readiness for ubiquitous adoption by 2030[33]. The expansion of DWPT technology also faces constraints related to capital and construction time.

Following the enactment of the Infrastructure Investment and Jobs Act in 2021, which accelerated road construction activities, approximately 111k kilometers of roadways have undergone refurbishment, equating to an average of 37k kilometers annually[34]. Incorporating DWPT systems concurrently with these road repairs could substantially reduce the infrastructure development timeline and costs. Based on the deployment scenarios modeled in this study, embedding DWPT in 103k lane-kilometers of interstates, 31k lane-kilometers of freeways, and 247k lane-kilometers of principal arterial roads would necessitate approximately 2.8, 0.84, and 9.1 years, respectively, culminating in a total minimum integration period of roughly 12.7 years. This timeline indicates that a nationwide rollout of DWPT could be achievable from 2030 to 2050, assuming the technology's readiness for widespread and rapid deployment by 2030, although this remains unlikely. Moreover, financing such a venture is capital-intensive, necessitating the utilization of funds from the Federal Highway Administration (FHWA), which possesses an annual budget of 70 billion USD, equating to a 20-year budget of 1.4 trillion USD[34]. It is estimated that 67 billion to 1.1 trillion USD would be required for civil costs every 20 years, while 67 billion to 610 billion USD in debt financing from the private sector would be needed for the electronics component over the same period. Consequently, the viability of garnering adequate financial resources for nationwide DWPT deployment hinges on the specific scenario.

**Table 1 | Vehicle operating ranges and battery sizes for medium-duty vehicles (MDVs) and heavy-duty vehicles (HDVs)**

| Primary operating range[42] | Operating distance | Portion of MDV VKT[42] | Portion of HDV VKT[42] | MDV battery size | HDV battery size | MDV portion public charging | HDV portion public charging |
|---|---|---|---|---|---|---|---|
| Off-road | 0 km | 0.89% | 2.7% | 90 kWh | 90 kWh | 0% | 0% |
| Under 80 km (50 mi) | 40 km | 79% | 23% | 90 kWh | 90 kWh | 0% | 0% |
| 81 to 161 km (50–100 mi) | 121 km | 13% | 14% | 180 kWh | 270 kWh | 0% | 0% |
| 162 to 322 km (101–200 mi) | 241 km | 3.7% | 12% | 180 kWh | 360 kWh | 12% | 11% |
| 323 to 805 km (201– 500 mi) | 563 km | 1.8% | 19% | 450 kWh | 810 kWh | 5.3% | 14% |
| 806 km (501 mi) or more | 889 km | 1.7% | 30% | 720 kWh | 990 kWh | 4.0% | 33% |

*km* kilometer, *mi* mile, *VKT* vehicle kilometers traveled, *kWh* kilowatt-hour.

If the deployment of DWPT is restricted to specific corridors, then it may necessitate EVs to utilize full-size batteries and multiple charging systems. Given that each charging system can be implemented simultaneously, their deployment can be optimized in areas where they offer the most cost-effective solution[35]. For instance, BSS is more advantageous in regions with high peak demand or time-of-use electricity price structures, whereas BSS may be less favorable in low-traffic areas due to expensive upfront capital costs. Moreover, each size of the BSS caters to different vehicle categories, with a small size for cars and LDTs and a large size for MDVs and HDVs. Thus, the deployment of BSS could be focused exclusively on MDVs and HDVs, which is especially advantageous due to their pressing charging requirements coupled with longer dwell times when utilizing DCFC and the ability for fleets to overcome social challenges with BSS[36]. DWPT eliminates charging dwell times, but has been shown to be most cost effective when utilized by all vehicle categories[12]. Ultimately, it is crucial to consider the TCO, GHG-intensity, and performance of each charging system when formulating planning and investment policies, as these factors are not equitably distributed across locations and vehicle categories.

## Methods

An integrated techno-economic analysis (TEA) and life cycle assessment (LCA) was developed to comprehensively compare the widescale deployment of DCFC, BSS, and DWPT charging infrastructure in the contiguous U.S. In this analysis, the three charging systems were deployed independently to facilitate comparison. The analysis focused on four vehicle categories: car, LDT, MDV, and HDV. For each of the vehicle categories, the TCO and cradle-to-grave GHG-intensity were evaluated for each of the EV charging systems and compared to a representative HEV and ICEV with a functional unit of one VKT. The analysis considered the implementation of charging infrastructure in 2030, aligning with the expected mass market adoption of EVs[4,20]. Infrastructure deployment was exclusively modeled in 2030 to ensure a consistent comparison across charging systems, followed by their 20-year operational period from 2031 to 2050[12,37].

### Charging system energy usage

The usage of public charging was evaluated for DCFC, BSS, and DWPT. The DWPT system was assumed to maintain the vehicle's state of charge while driving on the electrified roadway, whereas the DCFC and BSS systems were modeled to provide energy only during daytime trips. The vehicle energy efficiencies per VKT were 0.19 kWh for cars, 0.30 kWh for LDTs, 0.68 kWh for MDVs, 1.34 kWh for HDVs, and 1.35 kWh for buses[38]. The usage of these systems was categorized into stationary charging usage (DCFC and BSS) and DWPT roadway usage. Cars, LDTs, MDVs, and HDVs were assumed to utilize the stationary charging systems and DWPT roads. Buses were included in the usage of the DWPT roadway; however, their TCO and GHG-intensity were not explicitly modeled in the analysis due to their minimal VKT[39].

For each vehicle category, the usage of stationary charging systems was modeled separately. Cars and LDTs were assumed to have the same usage since they are both considered light-duty vehicles. Among light-duty vehicles, only battery electric vehicles were assumed to utilize the infrastructure, as plug-in hybrid electric vehicles are typically incompatible with the high-power rates of DCFC and the standardized battery sizes required for BSS. Hence, it was estimated that 82% of electric cars and LDTs would use the infrastructure[40]. Observational data indicated that public charging usage for light-duty vehicles is around 6%, resulting in a 5% usage of public charging for electric cars and LDTs[41].

However, due to the lack of available data for electric MDVs and HDVs, the usage of public charging was simulated for multiple vehicle operating ranges and corresponding battery sizes[42]. MDVs and HDVs were assumed to undergo overnight charging and start each day with a fully charged battery[29]. The battery sizes were divided into 90-kWh battery modules[43], and the number of modules onboard the EV was determined to minimize battery expenses while ensuring that the vehicle required no more than one public charging event per day during the operator's required driving break[44]. The simulated driving break was modeled such that the vehicle's state of charge would be above 20% before the charging event to maintain battery health and below 80% at the end of the charging event to optimize charging time[45]. The vehicle was assumed to only charge the minimum amount to complete its trip. The battery size, portion of VKT in the vehicle category, and portion of public charging usage are presented in Table 1 for each operating range. The average portion of energy supplied from public charging, weighted by VKT in each operating range, was found to be 0.6% for MDVs and 14% for HDVs.

The electrified roadway was assumed to provide continuous power, maintain the vehicle's state of charge, and be used by cars, LDTs, MDVs, HDVs, and buses. The minimum portion ($R$) of each roadway segment ($i$) that is needed to be electrified for each vehicle category was calculated in Eq. (2) based on the segment's speed limit ($S$), receiving pad power rating ($P$) of 50-kW, number of receiving pads ($N$) on the vehicle ($v$), vehicle energy efficiencies per VKT ($EE$), charging efficiency ($CE$) of 85%[46], and amount failed ($F$). The amount failed was calculated in Eq. (1) based on the VKT by each vehicle type over the life of the roadway segment, number of receiving pads on the vehicle, failure rate ($FR$) of 2.87 pads per million hours[47], number of roadway pads ($RP$) per segment (200 per kilometer (km))[12], and speed limit.

$$F_i = \sum_v FR*VKT_{i,v}*N_v/(RP_i*S_i) \qquad (1)$$

$$R_{v,i} = S_i*EE_v/(P*N_v*CE*(1 - F_i)) \qquad (2)$$

The electrified portion of the roadway was modeled based on the vehicle category that needed the highest portion electrified. The number of receiving pads for each vehicle category depended on their

charging requirements and wheelbase allowances, with cars having one pad, LDTs having two pads, MDVs having four pads, and HDVs and buses having five pads[12].

## Time of day usage

Time-of-day resolution was added to the EV energy demand using arrival and departure time data for cars, LDTs, MDVs, and HDVs. The charging schedule for DWPT and BSS was aligned with the vehicles' in-route periods since these systems have minimal charging times (Supplementary Fig. 6). In contrast, the DCFC schedule corresponded to the vehicles' dwell periods due to the slower charging rate of DCFC (Supplementary Fig. 7).

The arrival and departure time data for cars and LDTs were extracted from the 2017 National Household Travel Survey, which provides trip data for various vehicles[48]. Specifically, the schedule for cars was derived from 280k automobile trips, while the LDT schedule was based on 289k van, sport utility vehicle, pickup truck, other truck, and recreational vehicle trips. The energy consumed during each trip was assumed to represent the amount of energy replenished through charging. For in-route charging, the trip energy was evenly distributed throughout the trip, calculated based on the trip distance and vehicle energy efficiency. For charging during car and LDT dwell periods, the energy was replenished up to the amount consumed during the trip.

The charging schedules for MDVs and HDVs were determined using the National Renewable Energy Laboratory's Fleet DNA database, which contains operating data for commercial fleet vehicles[49]. The MDV schedule was developed from 1471 trips made by class 3 to 7 delivery trucks and vans, while the HDV schedule was based on 969 trips made by class 8 tractors. The trip data was categorized into each vehicle operating range and weighted by VKT from Table 1; trip data for over 322 VKT (200 vehicle miles travelled) was used for all operating ranges above 322 km. It was assumed that the VKT for each trip was evenly distributed between the arrival and departure times, resulting in a distribution of the in-route charging profile throughout the day. The weighted average charging schedules for MDVs and HDVs are illustrated in Supplementary Figs. 6 and 7.

## Deployment of infrastructure

Deployment scenarios were developed for DCFC, BSS, and DWPT to assess the charging cost and GHG-intensity of individual charging locations across the U.S. The estimated usage of each charging system was scaled using yearly traffic data and EV adoption projections. Vehicle traffic data from the Freight Analysis Framework Version 4 (FAF4) provided VKT estimates for 2012 and 2045 on 663k individual roadways for freight trucks (MDV and HDV) and all vehicles[19]. These estimates were interpolated and extrapolated linearly up to 2050.

To break down the FAF4 data by vehicle category, multiple datasets from the FHWA were utilized, incorporating the 2019 FHWA VKT data and their projected increase in 2049[24]. The FHWA VKT data included breakdowns for cars, LDTs, single-unit trucks (MDVs), combination trucks (HDVs), motorcycles, and buses[39,50]. The FHWA VKT data were used to calculate the portion of vehicles in the FAF4 data that fell into the categories of cars, LDTs, MDVs, and HDVs based on the percentage of FHWA VKT from each vehicle category on state roadways, including interstates (FAF4 interstates), other arterials (FAF4 freeways, principal arterials, and minor arterials), and other road types (FAF4 major collector and minor collector)[39]. It is worth noting that the FAF4 data did not include VKT on local roads. As a result, the FAF4 VKT data for stationary charging systems (DCFC and BSS) were scaled to match the total VKT of the FHWA data for each vehicle category in 2019 and 2049. In contrast, the FAF4 data for DWPT were not scaled to match the FHWA VKT total, as the VKT on individual roads was directly used in the analysis.

The VKT data, categorized by vehicle type on each roadway segment, were combined with EV adoption forecasts (Supplementary Figs. 1 and 2), a charging efficiency of 85% for each system[38,46], and vehicle energy efficiencies to estimate the charging demand from EVs. To account for uncertainty, three EV adoption scenarios were considered: optimistic, baseline, and conservative. The optimistic adoption curves for MDVs and HDVs were derived from Konstantinou and Gkritza (2023)[4], while the conservative and optimistic scenarios for cars, LDTs, and buses, as well as the conservative and baseline scenarios for MDVs and HDVs, were obtained from Mai et al.[20]. The baseline scenario for cars, LDTs, and buses represented the average of the conservative and optimistic EV adoption rates.

In summary, the energy demand for public EV charging was computed on major roadways in the contiguous U.S. from 2031 to 2050. The energy demand included hourly and yearly resolution for cars, LDTs, MDVs, HDVs, and buses.

The yearly energy demand on the roadways from each EV adoption scenario was used to allocate EV charging to suitable charging sites. A total of 122k potential charging site locations were identified, including 85k gas stations[51], 30k public surface parking lots[51], and 7k existing DCFC sites[3]. The suitability of the sites for high-power charging stations was evaluated based on their proximity to grid interconnections and minimum EV charging utilization. Since load growth from widescale EV adoption was expected to require new substations[23], the location of grid interconnections was modeled as transmission lines with voltages under 200-kV rather than existing substations[52]. Site locations within 6-km of grid interconnections[53], based on the 95th percentile of existing DCFC sites, were deemed to be within the maximum proximity to grid interconnections. Further, sites within 1.5-km were not restricted on their maximum power due to the allowances of line extension policy[54]. Sites with power limitations were restricted to a maximum power of 2.5 megawatts[55].

DCFC sites without power limitations were restricted to a maximum daily energy dispensed of 30% of the time[56] for 32 stations with space constraints, as observed. The DCFC stations were set to use either 150-kW or 350-kW chargers as observed from major charging networks[57]. In contrast, BSS energy dispensation was constrained by a 3-min swap time[10], limiting the maximum number of swaps during peak demand hours to prevent queuing. Each BSS site was designed to have two sizes of swapping stations: a small size for cars and LDTs, and a large size for MDVs and HDVs.

The maximum capacity ($m1$) of each site ($j$) was used in a gravity model (Eq. (4)) to allocate the yearly demand for EV charging on each roadway segment to the nearest 30 sites[58]. The allocation of EV charging ($F$) to each site was also based on the amount of charging demand on the roadways ($m2$), the distance between the charging site and roadway ($d$), and a scalar ($g$). The scalar $g$ was computed in Eq. (3) to ensure that the sum of $F$ was equal to $m2$ for each roadway segment.

$$g = \left[\sum_{1}^{30} m1_j/d_j^2\right]^{-1} \tag{3}$$

$$F_j = g*m1_j*m2/d_j^2 \tag{4}$$

The allocation of EV charging to each site was then corrected to ensure that the maximum capacity of the site was not exceeded, and sites with very low usage were removed to avoid poor economics. The minimum allowed energy allocated to a DCFC site was set such that one 150-kW charger would dispense energy at least 5% of the time based on today's conditions[57], which represented the minimum usage threshold for the highest demand year. Similarly, each BSS site was designed to have a total energy demand of at least 4 swaps per day for both sizes of swapping stations during the highest demand year. Equations (3–4) were then used to allocate the energy for the remaining viable locations.

## Table 2 | Parameters used for charging systems and vehicles

| Parameter | Value | Units |
|---|---|---|
| **Direct Current Fast Charging** | | |
| 150-kW procurement | (a) 103k, (b) 119k, (c) 136k[57] | USD/charger |
| 350-kW procurement | (a) 174k, (b) 189k, (c) 204k[57] | USD/charger |
| 150-kW installation (chargers/site) | 59k (1), 47k (2), 35k (3–5), 23k (6+)[55] | USD/charger |
| 350-kW installation (chargers/site) | 82k (1), 65k (2), 48k (3–5), 32k (6+)[55] | USD/charger |
| Maintenance | 5% of procurement[84] | %/charger-year |
| Network contract | 229[57] | USD/charger-year |
| Data contract | 165[57] | USD/charger-year |
| **Battery Swapping** | | |
| 7.7-kW procurement | (a) 3.4k, (b) 3.7k, (c) 4.1k[57] | USD/charger |
| 50-kW procurement | (a) 27k, (b) 38k, (c) 49k[57] | USD/charger |
| 7.7-kW installation | 2.9k[55] | USD/charger |
| 50-kW installation | 22k[55] | USD/charger |
| Battery cabinet | 175[85] | USD/kWh-nameplate |
| Automated storage and retrieval system | (a) 48k, (b) 95k[86], (c) 143k | USD/BSS |
| Building | 3.1k[87] | USD/square-meter |
| Maintenance | 5% of procurement[84] | %/charger-year |
| **Dynamic Wireless Power Transfer** | | |
| Rural implementation | (a) 0.94M[11], (b) 2.6M[12,23], (c) 3.9M[23] | USD/lane-km |
| Urban implementation | (a) 2.2M[11], (b) 4.2M[12,23], (c) 5.4M[12,23] | USD/lane-km |
| Replacement inverter | 487k[23] | USD/km |
| **Total cost of ownership** | | |
| EV price without battery | 25k (car), 32k (LDT), 100k (MDV), 150k (HDV)[21] | USD/vehicle |
| EV battery price | (a) 90, (b) 118, (c) 150[22] | USD/kWh-nameplate |
| EV battery full-size | 70 (car)[72], 112 (LDT), 122 (MDV), 549 (HDV) | kWh/vehicle |
| EV battery short range | 13 (car), 21 (LDT), 48 (MDV), 94 (HDV) | kWh/vehicle |
| HEV selling price | 36k (car), 51k (LDT), 101k (MDV), 190k (HDV)[21] | USD/vehicle |
| ICEV selling price | 33k (car), 45k (LDT), 93k (MDV), 179k (HDV)[21] | USD/vehicle |
| EV maintenance | 0.07 (car), 0.08 (LDT), 0.10 (MDV), 0.11 (HDV)[88] | USD/VKT |
| HEV maintenance | 0.10 (car), 0.11 (LDT), 0.11 (MDV), 0.12 (HDV)[88] | USD/VKT |
| ICEV maintenance | 0.11 (car), 0.15 (LDT), 0.15 (MDV), 0.13 (HDV)[88] | USD/VKT |
| Annual VKT | 20k (car), 17k (LDT), 20k (MDV), 96k (HDV)[77] | VKT/vehicle-year |
| ICEV fuel economy | 51 (car), 36 (LDT), 11 (MDV), 13 (HDV)[38] | VKT/gallon |
| HEV fuel economy | 77 (car), 52 (LDT), 17 (MDV), 14 (HDV)[38] | VKT/gallon |

Capital and operational costs for Direct Current Fast Charging, Battery Swapping (BSS), and Dynamic Wireless Power Transfer systems, along with total cost of ownership parameters for the modeled passenger car (car), light-duty truck (LDT), medium-duty vehicle (MDV), and heavy-duty vehicle (HDV) as well as electric vehicle (EV), hybrid electric vehicle (HEV), and internal combustion engine vehicle (ICEV).
The table shows values for (a) optimistic, (b) baseline, and (c) conservative scenarios in 2022 United States Dollars (USD).
k thousand, M million, kW kilowatt, kWh kilowatt-hour, VKT vehicle kilometers traveled.

The required amount of charging equipment at each site was then determined based on its expected usage. For DCFC sites, the number of chargers needed was calculated separately for the first (2031–2040) and second (2041–2050) 10-year life of the equipment[37]. DCFC sites with low expected usage were equipped with 150-kW chargers, whereas those with high expected usage were equipped with 350-kW chargers. Specifically, 150-kW chargers were only deployed at sites where the highest usage was below the maximum capacity of a single 350-kW charger during the initial 10-year period, and also below the maximum combined spatial capacity of 32 150-kW chargers over the full 20-year period. Alternatively, for BSS sites, the number of batteries and support equipment needed was determined annually, with a minimum usage of 4 batteries per site. Supplementary Fig. 3 shows the coverage of DCFC and BSS infrastructure within 80 km (50 miles[1]) for each EV adoption scenario.

DWPT infrastructure was deployed on major roadways in the contiguous U.S. to maintain every vehicle's state of charge. The energy dispensed from the DWPT roadway was set to match the vehicle energy consumption on each roadway. If over half of vehicle traffic saturates the DWPT lane, a second lane is assumed to be electrified in each direction, reducing the utilization by half for the analyzed lane.

### Techno-economic analysis

The TEA conducted an evaluation of the charging cost and TCO for EVs using DCFC, BSS, and DWPT charging systems. In this study, only public charging costs from these systems were considered for the TCO comparison, although the actual TCO would include a mix of charging costs from home, workplace, fleet, public, and other locations. To capture the full range of values, optimistic, baseline, and conservative scenarios were developed for capital costs, electricity prices (Supplementary Fig. 4), and EV adoption (Supplementary Figs. 1 and 2), resulting in 27 charging cost and TCO scenarios for EVs. In addition, optimistic, baseline, and conservative scenarios were developed for traditional fuel prices (Supplementary Fig. 5) to evaluate refueling costs for ICEVs and HEVs.

The charging cost for each DCFC site, BSS site, and DWPT roadway segment was evaluated individually using a discounted cash flow rate of return (DCFROR). The DCFROR considered capital costs, operational costs, electricity costs, and utilization. The DCFROR assumed a 5% internal rate of return, capital debt financing of 50% with 6% interest and 10-year loan term, state and average local sales tax (Supplementary Table 1)[59], corporate federal (21%), and state income tax (Supplementary Table 1)[60], and a modified accelerated cost recovery system depreciation schedule. The cash flow spanned 21 years, including a 1-year build period and a 20-year operating life (2031 to 2050). The charging cost was calculated such that a net present value of zero was achieved. All costs were converted to 2022 USD using consumer price indexes[61,62] and producer price indexes[63–66]. A summary of the costs for each charging system and vehicle is presented in Table 2.

The capital costs for DCFC, BSS, and DWPT were evaluated individually and listed in Table 2. For every system, it was assumed that utilities would cover the cost of substations and line-extensions up to a certain distance[54], with expenses being recouped through electricity sales. As noted in Nelder and Rogers (2019), however, certain utilities might have imposed line-extension fees[57]. Further, this study assumed that the land of each site was owned already and did not depreciate. Therefore, capital costs were limited to the installation and procurement of all necessary components of each charging system.

The capital costs for DCFC were calculated by scaling the costs (Table 2) with the number of chargers at the site. These costs included a procurement component that was scaled linearly and an installation component that decreased on a per charger basis as the number increased. The procurement cost was incurred in 2030 and 2040, corresponding to the number of chargers deployed during each

respective 10-year period. The installation costs, on the other hand, were incurred upfront in 2030 to future-proof the charging system for both sets of charger lifespans[37].

In contrast, the capital costs for BSS (Table 2) included fixed costs for the small (19 square meters) and large (46 square meters) sizes of BSSs, which covered the automated storage and retrieval system required to swap batteries and the building housing the system. The number of cabinets, comprising containers, thermal management systems, and fire suppression systems, were determined based on the number of batteries needed to meet the annual demand. Furthermore, the number of chargers for each BSS was calculated based on the maximum charging load from 2031 to 2040 and from 2041 to 2050, utilizing 7.7-kW chargers for the small BSS and 50-kW chargers for the large BSS. Similar to DCFC, the procurement costs for the chargers were incurred in 2030 and 2040 for the first and second set of chargers required, respectively. The installation, automated storage and retrieval system, and building costs were incurred in 2030, while the battery and cabinet costs were incurred in the respective years when they were added to the BSS.

For DWPT, the capital cost (Table 2) was scaled according to the electrified distance of each roadway. Separate estimates were utilized for urban and rural roads, taking into account the substantial difference in civil costs between the two[67]. The electronics cost was assumed to be the same for both urban and rural roads. For the optimistic scenarios, the low estimate from Limb et al. (2019) was used for rural roads, while the high estimate was used for urban roads[11]. As for the baseline and conservative scenarios, the electronics cost of 1.6 million USD per km (adjusted to 2022 USD[63]), as reported by Haddad et al. (2022), was employed[23]. The conservative civil cost for urban roads was derived from the 1st-of-a-kind case in Trinko et al. (2022), with a lower contingency cost of 10% compared to the original 30%[12]. The baseline urban civil cost was also adapted from Trinko et al. (2022), incorporating a combination of the 1st-of-kind and nth-of-a-kind cases, which is further detailed in Supplementary Table 2[12]. Regarding the civil costs for rural roads, the nth-of-kind case from Trinko et al. (2022) was utilized for the baseline scenario, while the estimate from Haddad et al. (2022) was adapted (without substation) for the conservative scenario[23].

The DCFC sites were modeled to have data contracts, network contracts, and maintenance costs annually (Table 2). BSS sites were modeled to only have maintenance costs (Table 2) on the chargers. The replacement of BSS batteries was assumed to be the responsibility of the vehicle owner. The operational costs for DWPT consisted of replacing failed roadside inverters (Table 2) with a mean time to failure of 101 years[68]. Since the failed DWPT roadway pads were modeled to have excess capacity in the design (Eq. (2)), they were not replaced. The maintenance costs of the roadway were assumed to be out of scope since they are typically paid for by taxes, which are not part of the DCFC and BSS analysis.

Commercial electricity schedules from the U.S. Utility Rate Database[69] were collected[70] for the largest utility company in each state to determine electricity costs for each DCFC site, BSS site, and DWPT roadway segment on an annual basis. The electricity schedules were categorized into demand charges (USD/kW·month), electricity rates (USD/kWh), and fixed charges (USD/month), with applicable demand charges and electricity rates determined by the time-of-day charging profiles (Supplementary Figs. 6 and 7). The fixed charges were assessed to each BSS and DCFC site as well as to each DWPT roadway segment per 16 lane-km of electrified road. The most affordable schedule was selected based on the service location and power range for each load, with BSS charging profiles optimized to minimize electricity costs.

The BSS charging profile was optimized to ensure that each battery could be fully charged prior to the swap, with a one-hour buffer period. The charging time (t) required to charge the battery of each

vehicle (v) was calculated using Eq. (5), which accounts for the charger rating (p) (7.7 kW for small BSS and 50 kW for large BSS), average power rate (a) (95%), charging efficiency (e) of 85%[38], starting state of charge ($soc_s$) (20%)[45], final state of charge ($soc_f$) (80%)[45], and battery size (b) (Table 2).

$$t_v = \left( soc_f - soc_s \right) * b_v / (a * p_v * e) \tag{5}$$

The minimum number of batteries needed was determined by considering the required charging time and the swap schedule of batteries within the BSS (Supplementary Fig. 6). These constraints were integrated into the charging load optimization algorithm, which aimed to minimize electricity costs while ensuring that each battery was charged within the designated window and that the daily charging volume met the demand from EVs.

Once the electricity costs were calculated for each charging system, the electricity costs were adjusted using the 2022 and 2031 to 2050 price projections for generation (electricity rate) and distribution (demand charge) from the Annual Energy Outlook (2023)[71]. Three scenarios were considered to account for future changes in electricity prices (Supplementary Fig. 4): optimistic, baseline, and conservative.

The TCO analysis utilized the charging cost results for DCFC, BSS, and DWPT to estimate the charging cost for EVs per VKT. The TCO of each modeled EV was compared to that of a HEV and an ICEV. The fuel prices for HEVs and ICEVs were broken out by state and adjusted to 2031 through 2050 values (Supplementary Fig. 5) for optimistic, baseline, and conservative scenarios[71].

The TCO was computed over the first 10 years of the vehicle's life and included the charging or fueling cost (Supplementary Table 3), depreciation (Table 2), maintenance (Table 2), license and registration (Supplementary Table 4), and insurance (Supplementary Table 5) expenses. The analysis of ICEVs and HEVs considered gasoline fuel for cars and LDTs, and diesel fuel for MDVs and HDVs. Cars and LDTs were assumed to have a discount factor of 1.2% for yearly expenses, while MDVs and HDVs had a discount factor of 3%[21].

The purchase price (Table 2) for each vehicle category was determined based on an average vehicle. For EVs, the purchase price included the cost of the EV without the battery and the marked-up cost of the EV battery (Table 2). The price of an EV without the battery was obtained from Burnham et al. (2021) for a 2025 model year midsize sedan (car), pickup truck (LDT), class 6 pickup/delivery truck (MDV), and sleeper tractor (HDV)[21]. The battery size for electric LDTs was adjusted to match the range of an electric car by considering vehicle efficiencies[72]. Reduced battery sizes for DWPT EVs were determined for a 56 km operating range with a maximum depth of discharge of 80%.

The vehicle depreciation cost was calculated annually based on the purchase price. Cars and LDTs lost 29% of their original value in the first year and 11% of their remaining value in each consecutive year[21]. For MDVs and HDVs, 9% of their remaining value was lost every year[21]. Insurance costs were assessed based on the remaining value of the vehicle each year and the location of the charging system. License and registration costs were fixed annually and varied by state. Maintenance costs were based on the U.S. average fixed rate per VKT plus any battery replacement costs. Battery life was assumed to be 1000 full cycles for full-size batteries[73], while reduced battery sizes charged via DWPT were assumed to have the same life in years due to optimal operating characteristics, such as a smaller depth of discharge and a state of charge that can be maintained around 50%[74–76]. Based on these assumptions, only electric HDVs needed battery replacements in the first 10-year period due to their high annual VKT[77]. The sum of the vehicle costs was discounted along with their yearly utilization to obtain the TCO on a per VKT basis.

The impact of switching from ICEVs to EVs charged with DCFC, BSS, or DWPT systems (cs) to the overall cost of on-road transportation

was calculated using Eq. (6) for the percentage change ($\Delta TCO\%$) and Eq. (7) for the change in USD ($\Delta TCO\$$). These equations used the TCO of an ICEV ($TCO_{ICEV}$) and an EV ($TCO_{EV}$) for each vehicle category ($v$), as well as the VKT of EVs ($VKT_{EV}$) and the VKT of all vehicle powertrains ($VKT_{All}$).

$$\triangle TCO\%_{CS} = 1 + \left(\left[\sum_v VKT_{EV,v}*(TCO_{EV,cs,v} - TCO_{ICEV,v})\right] \Big/ \left[\sum_v VKT_{All,v}*TCO_{ICEV,v}\right]\right) \quad (6)$$

$$\triangle TCO_{CS} = \sum_v VKT_{EV,v}*(TCO_{EV,cs,v} - TCO_{ICEV,v}) \quad (7)$$

## Life cycle assessment

An attributional LCA was conducted to compare the GHG-intensity of EVs charged with DCFC, BSS, and DWPT, as well as ICEVs and HEVs. Specifically, the impact assessment used an economic allocation method and the 100-year global warming potential from the Intergovernmental Panel on Climate Change's (IPCC) 6th impact assessment report[78]. The study used a cradle-to-grave system boundary with a functional unit of one VKT. For EVs, the emissions were divided into charging emissions, embodied charging infrastructure emissions, and embodied vehicle emissions for different vehicle categories: cars, LDTs, MDVs, and HDVs. Charging and infrastructure emissions were allocated on a per unit of energy dispensed basis (kWh). The study used Ecoinvent 3.8 and openLCA 3.10 to collect the life cycle inventory data for charging infrastructure and charging emissions[79]. The Greenhouse gases, Regulated Emissions, and Energy use in Technologies model (GREET) 2022 was used to determine embodied vehicle emissions, as well as the HEV and ICEV feedstock, fuel, and vehicle operation emissions[38]. The feedstock and fuel emissions were combined as the equivalent infrastructure emissions for HEVs and ICEVs.

The DCFC infrastructure emissions were calculated based on the charger pedestal, power cabinet, implementation, and construction (Supplementary Table 6)[16]. The pedestal inventory data were taken from Ecoinvent 3.8 and scaled to a weight of 250-kg for 150-kW and 350-kW chargers[79,80]. Emissions from the power cabinet were broken out by material (Supplementary Table 7) using primary data for a weight of 1340-kg per 150-kW charger and 2680-kg per 350-kW charger[80]. Implementation emissions for DCFC were adapted from Lucas et al.[81].

The BSS infrastructure emissions included the charger pedestal, battery, battery cabinet, automated storage and retrieval system,

building, and construction (Supplementary Table 8). The emissions of each BSS site were scaled based on the amount of equipment used.

The DWPT infrastructure emissions were based on the electronics, pavement, and construction (Supplementary Table 9). DWPT infrastructure components were based on Marmiroli et al. (2019), however, the emissions factors were adjusted to use concrete rather than asphalt and use the IPCC 6th impact assessment[17]. The emissions for DWPT were scaled based on the electrified distance.

The emissions from EV charging were calculated using the forecasted hourly electricity mix in 134 Cambium (2022) zones from 2031 through 2050[82]. Three electricity mix scenarios were examined: optimistic, based on the Cambium (2022) mid-case with 100% decarbonization by 2035; baseline, based on the Cambium (2022) mid-case; and conservative, based on the Cambium (2022) high renewable energy cost. The charging emissions were calculated using the average electricity consumption mix from each zone, rather than the marginal mix[83].

The consumption mix was determined by tracking the net energy imports and exports from each zone within the Western, Eastern, and Texas U.S. interconnections. Furthermore, the electricity mix used for charging energy storage resources was accounted for and assigned to the mix at the time of EV charging. Emissions factors for various grid resources in North American Reliability Corporation regions were obtained from Ecoinvent 3.8, encompassing both operating and embodied emissions (Supplementary Table 10)[79]. Notably, carbon capture associated with the electricity grid was not attributed to EV charging as it was beyond the system boundary.

The charging emissions ($ChgGHG$) were computed on a per unit of energy basis (kWh) using Eq. (8). The calculation was based on each grid resource's ($r$) emissions factor ($EF$) and fraction of the consumption mix ($M$) at the time-of-day ($h$), year ($y$), and location ($z$) of charging.

$$ChgGHG_{h,y,z} = \sum_r EF_z * M_{r,h,y,z} \quad (8)$$

The charging emissions were then scaled (Eq. (9)) by the hourly and yearly charging load ($load$) from each DCFC site, BSS site, and DWPT road segment to get the average charging emissions per unit of energy (kWh) over the life of each system.

$$ChgGHG_z = \left[\sum_{y=1}^{20}\sum_{h=1}^{24} ChgGHG_{h,y,z}*load_{h,y,z}\right] \Big/ \left[\sum_{y=1}^{20}\sum_{h=1}^{24} load_{h,y,z}\right] \quad (9)$$

## Table 3 | Breakdown of vehicle emissions

| Parameter | Car[38] | LDT[38] | MDV[38] | HDV[38] |
|---|---|---|---|---|
| **Embodied vehicle emissions** | | | | |
| EV full-size battery | 5.7 tCO$_{2e}$/battery-life | 9.0 tCO$_{2e}$/battery-life | 11 tCO$_{2e}$/battery-life | 37 tCO$_{2e}$/battery-life |
| EV reduced battery | 1.0 tCO$_{2e}$/battery-life | 1.7 tCO$_{2e}$/battery-life | 5.3 tCO$_{2e}$/battery-life | 9.0 tCO$_{2e}$/battery-life |
| EV other components | 5.0 tCO$_{2e}$/life | 8.4 tCO$_{2e}$/life | 34 tCO$_{2e}$/life | 74 tCO$_{2e}$/life |
| HEV | 6.1 tCO$_{2e}$/life | 10 tCO$_{2e}$/life | 38 tCO$_{2e}$/life | 83 tCO$_{2e}$/life |
| ICEV | 6.2 tCO$_{2e}$/life | 10 tCO$_{2e}$/life | 37 tCO$_{2e}$/life | 81 tCO$_{2e}$/life |
| **Feedstock and fuel emissions** | | | | |
| HEV | 26 g-CO$_{2e}$/VKT | 39 g-CO$_{2e}$/VKT | 123 g-CO$_{2e}$/VKT | 156 g-CO$_{2e}$/VKT |
| ICEV | 40 g-CO$_{2e}$/VKT | 57 g-CO$_{2e}$/VKT | 191 g-CO$_{2e}$/VKT | 164 g-CO$_{2e}$/VKT |
| **Operating emissions** | | | | |
| HEV | 112 g-CO$_{2e}$/VKT | 166 g-CO$_{2e}$/VKT | 597 g-CO$_{2e}$/VKT | 757 g-CO$_{2e}$/VKT |
| ICEV | 171 g-CO$_{2e}$/VKT | 243 g-CO$_{2e}$/VKT | 924 g-CO$_{2e}$/VKT | 793 g-CO$_{2e}$/VKT |

The global warming potential is given for multiple vehicle types: electric vehicles (EV), hybrid electric vehicles (HEVs), and internal combustion engine vehicles (ICEVs); and vehicle categories: car, light-duty truck (LDT), medium-duty vehicle (MDV), and heavy-duty vehicle (HDV).
$CO_{2e}$ carbon dioxide equivalent, $t$ metric tonne, $g$ gram, VKT vehicle kilometer traveled.

Embodied vehicle emissions for each vehicle type were calculated using a representative 2025 vehicle with conventional materials from GREET (2022)[38]. Specifically, the vehicle modeled for each vehicle category from GREET (2022) were a passenger car, pickup truck (LDT), class 6 pickup-and-delivery truck (MDV), and class 8 sleeper-cab truck (HDV).

The vehicle emissions were divided into components; assembly, disposal, and recycling (ADR); batteries; and fluids. EV battery sizes (Table 2) were input into GREET (2022) for the corresponding charging system and EV-adoption scenario[38]. The batteries for both EVs and HEVs were assumed to be manufactured in China and use a lithium-ion chemistry. One replacement of the hybrid electric HDV battery was assumed to occur over the vehicle's lifetime. Further, electric MDVs and HDVs were calculated to average 1.7 and 2.9 battery replacements in their lifetime, while cars and LDTs had none. The modeled vehicle emissions are summarized in Table 3.

## Data availability
The figure data generated in this study are provided in the Source data file. All figures and Source data are available in the figshare repository: https://doi.org/10.6084/m9.figshare.23902368. Source data are provided with this paper.

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

## Acknowledgements

The authors are grateful for financial support from the National Science Foundation [grant number 1941524] for N.H., D.A.T., and J.C.Q. Map data from OpenStreetMap (openstreetmap.org/copyright).

## Author contributions

N.H. and J.C.Q. designed the research. N.H. and D.A.T. acquired and analyzed the data. N.H., D.A.T., and J.C.Q. contributed to writing the paper.

## Competing interests
The authors declare no competing interests.
