## [Peer Review File · Nature Communications]

REVIEWER COMMENTS

Reviewer #1 (Remarks to the Author):

This paper presents an estimate of total cost of ownership and greenhouse gas emissions resulting from deployment of various EV-charging infrastructure scenarios in the contiguous US, projected for the period 2031-2050. The work addresses several major emerging concerns with EV charging rollout including capital costs, EV usage differences across geographies/urban types, life-cycle climate impacts. The analysis also includes implications for light, medium, and heavy duty vehicles. The study is very comprehensive and methods and results are both clearly communicated. I recommend publishing, with the following critiques being addressed:

o The study presents TCO and CO₂e projections on a per-km-traveled basis at county-level resolution. Some clarifications are needed to fully contextualize these results:

A) Are there any differences in total km-traveled across scenarios? If not, then the per-km TCO and CO₂ values presented in the Figs are relatively the same across scenarios, scaled up to the entire population, however it would still be good to have the total annual emissions in 2050 presented, along with some evaluation of the % change relative to 2022. This is always useful in any paper doing a GHG evaluation as it gives some idea of alignment with Paris Climate targets (whether US is signatory or not). Similarly, how does the 10-year TCO or per-km TCO compare to today's values?

B) If no differences in km-traveled across scenarios, why not? Presumably differences in cost as well as technological configuration will drive behavioral change (via rebound effects, transportation mode choice shifting, economic growth, and societal change) across a 20-year time period. This should at least be acknowledged, if no attempt to model shifts in usage patterns.

o It is unclear where capital cost figures into the TCO values presented in Fig 2. Since capital cost is an upfront concern, esp for BSS and DWPT which are drastically new/infrastructure-heavy directions, it would be helpful to pull out capital cost entirely in this depiction.

o Please address or summarize the literature on the true feasibility of rolling out BSS and/or DWPT technologies in the coming decade, i.e. in time to make a 2031-2040 forecast a realistic consideration? It seems pretty well-established that these infrastructures cannot/will not be in place at any significant scale by then. I remember BSS being a prominent idea for EV Infrastructure rollout in 2005 and has made very little (if not negative) progress in 2 decades.

o I recommend using "GHG-intensity" to describe your CO₂e/km-traveled metric, rather than Global Warming Potential. In the climate impacts world, GWP is a strictly-defined term that describes the relative radiative forcing potential of various greenhouse gases.

Reviewer #3 (Remarks to the Author):

This study compares costs and climate impacts of various electric vehicle charging systems across the United States.

The work flow of the paper should be modified in a readable manner like Introduction, Literature review, problem formulation, methodology and results & discussion.

I could not see any research part/ novelty of the study/ contributions of the work.

This work may be considered as a case study not a research paper.

No clear explanation in the results part.

Reviewer #1 (Remarks to the Author):

This paper presents an estimate of total cost of ownership and greenhouse gas emissions resulting from deployment of various EV-charging infrastructure scenarios in the contiguous US, projected for the period 2031-2050. The work addresses several major emerging concerns with EV charging rollout including capital costs, EV usage differences across geographies/urban types, life-cycle climate impacts. The analysis also includes implications for light, medium, and heavy duty vehicles. The study is very comprehensive and methods and results are both clearly communicated. I recommend publishing, with the following critiques being addressed:

The authors are grateful to the reviewer for support and the constructive comments. We have diligently worked to address all of the comments and feel the manuscript has been improved through the revision process.

o The study presents TCO and CO₂e projections on a per-km-traveled basis at county-level resolution. Some clarifications are needed to fully contextualize these results:

A) Are there any differences in total km-traveled across scenarios? If not, then the per-km TCO and CO₂ values presented in the Figs are relatively the same across scenarios, scaled up to the entire population, however it would still be good to have the total annual emissions in 2050 presented, along with some evaluation of the % change relative to 2022. This is always useful in any paper doing a GHG evaluation as it gives some idea of alignment with Paris Climate targets (whether US is signatory or not).

Thank you for your suggestion. We have added text in the manuscript stating the total annual emissions in 2050 compared to 2022 and a supplementary figure. Please see the updated text and supplementary figure below addressing both parts of the comment:

“Taking into account an overall increase in VKT²⁴, which remains consistent across scenarios without considering changes in user behavior due to cost or technology shifts, the reduction in U.S. GHG emissions by 2050, relative to the baseline year of 2022, is estimated to span from 1% to 58% depending on the scenario (Supplementary Figure 11). This highlights the intricate challenges associated with emission reduction efforts amidst potential rises in vehicle usage.”

Supplementary Figure 11. Breakdown of the total vehicle greenhouse gas emissions in 2022 and 2050. The emissions are presented for optimistic (Opt.), baseline (Base.), and conservative (Cons.) electric vehicle adoption (Adopt) and electricity mix (Mix) scenarios. Vehicle emissions from the contiguous United States (U.S.) are shown for electric vehicles (EVs) charged via Direct Current Fast Charging (DCFC), Battery Swapping (BSS), and Dynamic Wireless Power Transfer (DWPT). Results are compared to an average internal combustion engine vehicle (ICEV) and hybrid electric vehicle (HEV). The results are broken down for passenger cars (Car), light-duty trucks (LDT), medium-duty vehicles (MDV), and heavy-duty vehicles (HDV).

Similarly, how does the 10-year TCO or per-km TCO compare to today's values?

Thank you for noting that a comparison to today's values will be of interest to the audience. Accordingly, we have included Supplementary Figure 10 to illustrate the current 10-year TCO, and it is now referenced in the Discussion section.

“Compared to 2022's TCO (Supplementary Figure 10)²¹, EV scenarios for 2031 to 2050 exhibit notably lower depreciation costs due to cheaper battery technologies but encounter higher charging costs from the premium associated with public charging infrastructure.”

Supplementary Figure 10. Breakdown of the 10-year total cost of ownership in 2022.

Results are presented for an average (A) passenger car, (B) light duty truck, (C) medium duty vehicle, and (D) heavy duty vehicle in the contiguous United States. The vehicle types include an average electric vehicle (EV), internal combustion engine vehicle (ICEV), and hybrid electric vehicle (HEV) from each vehicle category.

B) If no differences in km-travelled across scenarios, why not? Presumably differences in cost as well as technological configuration will drive behavioral change (via rebound effects, transportation mode choice shifting, economic growth, and societal change) across a 20-year time period. This should at least be acknowledged, if no attempt to model shifts in usage patterns.

Thank you for identifying that the total kilometers travelled from each scenario was not clearly articulated. While the total kilometers travelled by electric vehicles differs across electric vehicle adoption scenarios, the overall vehicle kilometers travelled for all vehicles remain constant regardless of technological configuration. We have accounted for the projected increase in

vehicle travel over time by the Federal Highway Administration. We added the following sentence to clarify the limitations of the vehicle kilometer travelled estimates in the revised manuscript:

“Taking into account an overall increase in VKT²⁴, which remains consistent across scenarios without considering changes in user behavior due to cost or technology shifts, the reduction in U.S. GHG emissions by 2050, relative to the baseline year of 2022, is estimated to span from 1% to 58% depending on the scenario (Supplementary Figure 11).”

o It is unclear where capital cost figures into the TCO values presented in Fig 2. Since capital cost is an upfront concern, esp for BSS and DWPT which are drastically new/infrastructure-heavy directions, it would be helpful to pull out capital cost entirely in this depiction.

Thank you for your insightful recommendation. In response, we have incorporated a hatch mark across all total cost of ownership figures to clearly delineate the capital cost component of charging expenses. Further, upon a diligent review of our manuscript, we identified an inadvertent discrepancy in the units of measurement presented in this figure. This has been promptly corrected to accurately display values per kilometer travelled, in lieu of the previously incorrect per mile travelled values. We have also conducted a thorough review of all other figures and text to confirm the accuracy of their units of measurement, which were verified to be correct.

Fig. 2. Breakdown of the 10-year total cost of ownership. Results are presented for an average (A) passenger car, (B) light duty truck, (C) medium duty vehicle, and (D) heavy duty vehicle in the contiguous United States. The vehicle types include electric vehicles charged via Direct Current Fast Charging (DCFC-EV), Battery Swapping (BSS-EV), and Dynamic Wireless Power Transfer (DWPT-EV). The EVs are compared to an average internal combustion engine vehicle (ICEV) and hybrid electric vehicle (HEV) from each vehicle category. The baseline scenarios are

shown in this static figure and all scenarios are shown in the interactive figure (see Supplementary Note 1) or repository.

o Please address or summarize the literature on the true feasibility of rolling out BSS and/or DWPT technologies in the coming decade, i.e. in time to make a 2031-2040 forecast a realistic consideration? It seems pretty well-established that these infrastructures cannot/will not be in place at any significant scale by then. I remember BSS being a prominent idea for EV Infrastructure rollout in 2005 and has made very little (if not negative) progress in 2 decades.

Thank you for highlighting this critical gap in our discussion. In response to your insightful feedback, we have incorporated the following text in the Discussion section addressing the practical feasibility of deploying BSS and DWPT technologies within the next decade:

“DCFC technology emerges as the most advanced in terms of readiness. DCFC's early adoption within the market was facilitated through financing from charging providers, automakers, and federal funds. Presently, DCFC maintains a substantial presence in the car and LDT sectors. However, less than 1% of DCFC charging stations are accessible to MDVs and HDVs³, presenting an opportunity for market disruption by BSS or DWPT.

Despite this potential, the widespread deployment of BSS and DWPT faces substantial challenges. BSS has seen limited deployment within the U.S. market but boasts over two thousand stations in China, predominantly installed by the automaker NIO³¹. This demonstrates the technological maturity of BSS, though with minimal market penetration. Further, the deployment of BSS in China for MDVs and HDVs showcases its adaptability to various vehicle requirements³². The early adoption of BSS in China has underscored the necessity for battery standardization—regarding size, shape, and attachment mechanisms—to facilitate efficient capital recovery and ensure vehicle compatibility⁵. Nonetheless, the reluctance of U.S. automakers to forego proprietary battery technologies and vehicle owners' resistance to battery sharing present significant social obstacles⁷. Consequently, the widespread acceptance of BSS in the U.S. appears improbable between 2030 and 2050 for cars and LDTs, with social barriers rather than technological readiness posing the primary hindrance.

Conversely, DWPT faces challenges related to its untested technological readiness and uncertain consumer receptivity. Electreon initiated the first on-road DWPT pilot in the U.S. in 2023, in Michigan, with further pilots by Magment GmbH and ENRX scheduled for 2024 in Indiana and Florida, respectively. Thus, the limited data from these deployments does not yet substantiate the reliability and general applicability of DWPT for widespread use. Additionally, the lack of standardized protocols for DWPT impedes its readiness for ubiquitous adoption by 2030³³. The expansion of DWPT technology also faces constraints related to capital and construction time.

Following the enactment of the Infrastructure Investment and Jobs Act in 2021, which accelerated road construction activities, approximately 111 thousand (k) kilometers of roadways have undergone refurbishment, equating to an average of 37k kilometers annually³⁴. Incorporating DWPT systems concurrently with these road repairs could substantially reduce the infrastructure development timeline and costs. Based on the deployment scenarios modeled in this study, embedding DWPT in 103k lane-kilometers of interstates, 31k lane-kilometers of

freeways, and 247k lane-kilometers of principal arterial roads would necessitate approximately 2.8, 0.84, and 9.1 years, respectively, culminating in a total minimum integration period of roughly 12.7 years. This timeline indicates that a nationwide rollout of DWPT could be achievable from 2030 to 2050, assuming the technology's readiness for widespread and rapid deployment by 2030, although this remains unlikely. Moreover, financing such a venture is capital-intensive, necessitating the utilization of funds from the Federal Highway Administration (FHWA), which possesses an annual budget of 70 billion USD, equating to a 20-year budget of 1.4 trillion USD³⁴. It is estimated that 67 billion to 1.1 trillion USD would be required for civil costs every 20 years, while 67 billion to 610 billion USD in debt financing from the private sector would be needed for the electronics component over the same period. Consequently, the viability of garnering adequate financial resources for nationwide DWPT deployment hinges on the specific scenario.”

o I recommend using "GHG-intensity" to describe your CO₂e/km-traveled metric, rather than Global Warming Potential. In the climate impacts world, GWP is a strictly-defined term that describes the relative radiative forcing potential of various greenhouse gases. Thank you for your suggestion. We have changed GWP to GHG-intensity or GHG emissions throughout the documents and figures.

Reviewer #3 (Remarks to the Author):

This study compares costs and climate impacts of various electric vehicle charging systems across the United States.

The authors appreciate the reviewer's valuable feedback. We have carefully addressed each comment, resulting in significant improvements to the manuscript through the revision process.

The work flow of the paper should be modified in a readable manner like Introduction, Literature review, problem formulation, methodology and results & discussion.

Thank you for your suggestion on reformatting the paper. The paper has been organized in the format required by Nature Communications: Abstract, Introduction, Results, Discussion, Methods. Following your suggestion, we have introduced headers for the Introduction, Results, Discussion, and Methods sections and enriched the Introduction with a literature review, restructuring it to transition from introduction to literature review, and then to problem formulation, as categorized below.

Introduction:

“The United States (U.S.) is currently undertaking an ambitious initiative to deploy public charging infrastructure to facilitate the widespread adoption of electric vehicles (EVs) necessary for achieving climate targets¹. As EVs continue to gain popularity in all vehicle classes, ensuring uninterrupted transportation has become a critical objective for policymakers and stakeholders^{1,2}. While initial efforts have focused on deploying Level 2 and Direct Current Fast Charging (DCFC) infrastructure³, a significant challenge lies in the charging time required to replenish EV batteries². Long charging times pose potential inconveniences for EV drivers, particularly those embarking on long journeys or requiring urgent charging². Addressing this issue necessitates the implementation of charging systems capable of fulfilling consumer needs^{2,4}. The electrification of medium-duty vehicles (MDVs) and heavy-duty vehicles (HDVs)⁴ changes the infrastructure needs to accommodate a broader range of vehicles beyond passenger cars and light-duty trucks (LDTs)⁵. Consequently, it is imperative to transition towards technologies that enable short dwell times for all vehicle types, such as 350-kilowatt (kW) DCFC, Battery Swapping (BSS), or Dynamic Wireless Power Transfer (DWPT)⁵.

Each of these technologies presents a distinct set of benefits and challenges. DCFC follows a similar model to traditional liquid refueling and allows for scalability by increasing the number of stations according to demand. However, the intermittent high-power loads of DCFC present challenges to the electrical grid and costs to the consumer⁶. In contrast, BSS optimizes grid loads by charging batteries before they are swapped, but their successful implementation relies on battery standardization and addressing social challenges related to battery ownership^{5,7}.

Additionally, BSS requires two different sizes: a small size for cars and LDTs and a large size for MDVs and HDVs⁸. Despite these challenges, BSS offers reduced dwell times compared to DCFC, making their dwell time comparable to that of internal combustion engine (ICEV) and hydrogen fuel cell vehicles^{9,10}. Alternatively, DWPT inductively charges vehicles while they are in motion using embedded electronics in the roadway, effectively enabling smaller batteries and

eliminating the need for vehicles to stop between destinations for recharging¹¹. DWPT, however, may cause traffic disruptions during roadway replacements, has limited deployment history, and is capital intensive^{12,13}. Despite the well-understood performance of these technologies, there remains a meaningful gap in understanding the economic and environmental implications that would arise from their widescale deployment.”

Literature Review:

“Current research on the economic and environmental impacts of EV charging systems is often narrowly focused, circumscribed by location, implementation scale, analysis scope, and scenario range. For instance, Wood et al. (2017) concentrated on the deployment of charging stations for varying levels of EV adoption without considering costs or greenhouse gas (GHG) emissions¹⁴. Muratori et al. (2019) examined variations in electricity costs for DCFC, finding utilization rates and locations meaningfully affect electricity costs but did not account for total charging costs with the exclusion of capital and operational expenses¹⁵. Mulrow & Grubert (2023) investigated the GHG emissions linked with traditional EV infrastructure, including DCFC, finding the embodied emissions of such infrastructure to be minimal¹⁶.

Conversely, the environmental impact of infrastructure emissions from DWPT and BSS remain unclear. Marmiroli et al. (2019) evaluated the GHG-intensity of DWPT infrastructure per distance, omitting an evaluation based on energy consumption¹⁷. Similarly, Limb et al. (2018) simulated the deployment of DWPT, analyzing charging costs and electricity emissions without considering the total cost of ownership (TCO) or lifecycle emissions¹¹. While BSS has been identified as potentially economically viable¹⁸, its broader economic and environmental assessments are lacking. In summary, although there exists a body of literature that partially addresses the costs and GHG emissions associated with EV charging systems, there has yet to be a comprehensive study that evaluates these aspects holistically, compares the three charging technologies of BSS, DCFC, and DWPT, or considers their implications across different vehicle categories.”

Problem Formulation:

“This study addresses this gap by simulating the nationwide deployment of DCFC, BSS, and DWPT and assessing the GHG-intensity and TCO of EVs utilizing these systems. By leveraging geospatially resolved charging demand, emissions, and cost data, this study determines location-specific sustainability outcomes. Deployment scenarios for the charging systems, spanning from 2031 to 2050, are formulated based on the geospatial demand derived from traffic data forecasts¹⁹ and three EV adoption scenarios (Supplementary Figs. 1-2): optimistic, baseline, and conservative^{4,20}. DCFC and BSS charging infrastructure is placed at existing DCFC sites, gas stations, and surface parking lots near grid interconnections. DCFC and BSS charging systems are modeled to represent a relatively small portion of EV charging, as the primary reliance remains on home or fleet charging. In contrast, DWPT infrastructure is deployed along major roadways to ensure that EVs can maintain their state of charge, consequently requiring a fixed amount of infrastructure per kilometer of roadway and providing a large portion of EV charging (see Methods)¹². Specifically, in optimistic and baseline EV adoption scenarios, DWPT is deployed on interstates, freeways, and principal arterial roads, thereby reducing the required EV battery sizes to a range of 56-kilometers (35-miles) (Supplementary Fig. 3)¹¹. However, with fewer EVs, the conservative EV adoption scenario assumes that only interstates are electrified, necessitating the use of full-size batteries for EVs. Simulation results are used to determine the

levelized cost of charging and GHG-intensity for each DCFC site, BSS site, and DWPT roadway, accounting for each location's design, utilization, electricity costs, and electricity mix. The levelized cost of charging is used to determine the TCO for EVs, which is then compared to ICEVs and hybrid electric vehicles (HEVs) over a 10-year period per vehicle-kilometer travelled (VKT)²¹. Finally, to account for the uncertainty of evolving variables that highly influence TCO and GHG-intensity, optimistic, baseline, and conservative scenarios are modeled for electricity mixes, capital costs, electricity prices, and fuel prices (Supplementary Figs. 4-5). These scenarios result in a total of 81 TCO and 9 GHG-intensity comparisons for each charging system with results presented as interactive figures. The findings reveal meaningful variability in TCO and GHG-intensity benefits of vehicle electrification, with impacts heavily dependent on the chosen charging system, location, vehicle category, and scenario, emphasizing the need for careful technology selection and innovation.”

I could not see any research part/ novelty of the study/ contributions of the work.

Thank you for identifying that the novelties and contributions of the study were not clearly stated. To address this, we have enhanced the Introduction and revised the Abstract. Below, the segments of the Abstract and Introduction that detail the identified research gaps, alongside the novelty and contributions of this study, are **highlighted**.

Abstract:

“The seamless adoption of electric vehicles (EVs) in the United States (U.S.) necessitates the development of extensive and effective charging infrastructure. Various charging systems have been proposed, including Direct Current Fast Charging (DCFC), Battery Swapping (BSS), and Dynamic Wireless Power Transfer (DWPT). **While many studies have evaluated the charging costs and greenhouse gas (GHG) intensity of EVs, a comprehensive analysis comparing these systems and their implications across vehicle categories remains unexplored.** This study compares the total cost of ownership (TCO) and GHG-intensity of EVs using these charging systems. Based on nationwide infrastructure deployment simulations, the change to TCO from adopting EVs varies by scenario, vehicle category, and location, with local fuel prices, electricity prices, and traffic volumes dramatically impacting results. Further, EV GHG-intensity depends on local electricity mixes and infrastructure utilizations. This research highlights the responsiveness of EV benefits resulting from technology advancements, deployment decisions, and policymaking.”

Introduction:

“Despite the well-understood performance of these technologies, there remains a meaningful gap in understanding the economic and environmental implications that would arise from their widescale deployment.

Current research on the economic and environmental impacts of EV charging systems is often narrowly focused, circumscribed by location, implementation scale, analysis scope, and scenario range. For instance, Wood et al. (2017) concentrated on the deployment of charging stations for

varying levels of EV adoption without considering costs or greenhouse gas (GHG) emissions¹⁴. Muratori et al. (2019) examined variations in electricity costs for DCFC, finding utilization rates and locations meaningfully affect electricity costs but did not account for total charging costs with the exclusion of capital and operational expenses¹⁵. Mulrow & Grubert (2023) investigated the GHG emissions linked with traditional EV infrastructure, including DCFC, finding the embodied emissions of such infrastructure to be minimal¹⁶.

Conversely, the environmental impact of infrastructure emissions from DWPT and BSS remain unclear. Marmioli et al. (2019) evaluated the GHG-intensity of DWPT infrastructure per distance, omitting an evaluation based on energy consumption¹⁷. Similarly, Limb et al. (2018) simulated the deployment of DWPT, analyzing charging costs and electricity emissions without considering the total cost of ownership (TCO) or lifecycle emissions¹¹. While BSS has been identified as potentially economically viable¹⁸, its broader economic and environmental assessments are lacking. In summary, although there exists a body of literature that partially addresses the costs and GHG emissions associated with EV charging systems, there has yet to be a comprehensive study that evaluates these aspects holistically, compares the three charging technologies of BSS, DCFC, and DWPT, or considers their implications across different vehicle categories.

This study addresses this gap by simulating the nationwide deployment of DCFC, BSS, and DWPT and assessing the GHG-intensity and TCO of EVs utilizing these systems.”

This work may be considered as a case study not a research paper.

We appreciate the reviewer’s diligence that this manuscript could be a case study rather than a research paper. However, we believe this manuscript exhibits the characteristics of a research paper. Specifically, this manuscript includes a broad examination of the implications of deploying public charging infrastructure for electric vehicles on a national scale. The manuscript is also composed of research objectives, such as assessing the greenhouse gas intensity and total cost of ownership of electric vehicles utilizing different charging systems. The manuscript engages in detailed data analysis, considering various factors like charging infrastructure deployment, electricity costs, emissions, and vehicle categories. The breakdowns of charging cost, total cost of ownership, and greenhouse gas intensity for different charging systems demonstrate a comprehensive and rigorous research methodology and represents a novel contribution to the research community. The inclusion of different scenarios (optimistic, baseline, conservative) and the consideration of evolving variables reflect a research-driven attempt to understand the potential outcomes under varying conditions. The manuscript concludes by summarizing the potential changes to transportation costs, greenhouse gas emissions, electricity grid infrastructure, and automotive manufacturing resulting from the transition to electric vehicles to provide a research-driven exploration of broader implications.

No clear explanation in the results part.

Thank you for noting that the results required further explanation. We have incorporated additional text and supplementary figures, and clearly broken down the capital costs for DCFC, BSS, and DWPT within the total cost of ownership figure:

“Compared to 2022's TCO (Supplementary Figure 10)²¹, EV scenarios for 2031 to 2050 exhibit notably lower depreciation costs due to cheaper battery technologies but encounter higher charging costs from the premium associated with public charging infrastructure. Taking into account an overall increase in VKT²⁴, which remains consistent across scenarios without considering changes in user behavior due to cost or technology shifts, the reduction in U.S. GHG emissions by 2050, relative to the baseline year of 2022, is estimated to span from 1% to 58% depending on the scenario (Supplementary Figure 11). This highlights the intricate challenges associated with emission reduction efforts amidst potential rises in vehicle usage.”

“DCFC technology emerges as the most advanced in terms of readiness. DCFC's early adoption within the market was facilitated through financing from charging providers, automakers, and federal funds. Presently, DCFC maintains a substantial presence in the car and LDT sectors. However, less than 1% of DCFC charging stations are accessible to MDVs and HDVs³, presenting an opportunity for market disruption by BSS or DWPT.

Despite this potential, the widespread deployment of BSS and DWPT faces substantial challenges. BSS has seen limited deployment within the U.S. market but boasts over two thousand stations in China, predominantly installed by the automaker NIO³¹. This demonstrates the technological maturity of BSS, though with minimal market penetration. Further, the deployment of BSS in China for MDVs and HDVs showcases its adaptability to various vehicle requirements³². The early adoption of BSS in China has underscored the necessity for battery standardization—regarding size, shape, and attachment mechanisms—to facilitate efficient capital recovery and ensure vehicle compatibility⁵. Nonetheless, the reluctance of U.S. automakers to forego proprietary battery technologies and vehicle owners' resistance to battery sharing present significant social obstacles⁷. Consequently, the widespread acceptance of BSS in the U.S. appears improbable between 2030 and 2050 for cars and LDTs, with social barriers rather than technological readiness posing the primary hindrance.

Conversely, DWPT faces challenges related to its untested technological readiness and uncertain consumer receptivity. Electreon initiated the first on-road DWPT pilot in the U.S. in 2023, in Michigan, with further pilots by Magment GmbH and ENRX scheduled for 2024 in Indiana and Florida, respectively. Thus, the limited data from these deployments does not yet substantiate the reliability and general applicability of DWPT for widespread use. Additionally, the lack of standardized protocols for DWPT impedes its readiness for ubiquitous adoption by 2030³³. The expansion of DWPT technology also faces constraints related to capital and construction time.

Following the enactment of the Infrastructure Investment and Jobs Act in 2021, which accelerated road construction activities, approximately 111 thousand (k) kilometers of roadways have undergone refurbishment, equating to an average of 37k kilometers annually³⁴. Incorporating DWPT systems concurrently with these road repairs could substantially reduce the infrastructure development timeline and costs. Based on the deployment scenarios modeled in this study, embedding DWPT in 103k lane-kilometers of interstates, 31k lane-kilometers of freeways, and 247k lane-kilometers of principal arterial roads would necessitate approximately 2.8, 0.84, and 9.1 years, respectively, culminating in a total minimum integration period of roughly 12.7 years. This timeline indicates that a nationwide rollout of DWPT could be

achievable from 2030 to 2050, assuming the technology's readiness for widespread and rapid deployment by 2030, although this remains unlikely. Moreover, financing such a venture is capital-intensive, necessitating the utilization of funds from the Federal Highway Administration (FHWA), which possesses an annual budget of 70 billion USD, equating to a 20-year budget of 1.4 trillion USD³⁴. It is estimated that 67 billion to 1.1 trillion USD would be required for civil costs every 20 years, while 67 billion to 610 billion USD in debt financing from the private sector would be needed for the electronics component over the same period. Consequently, the viability of garnering adequate financial resources for nationwide DWPT deployment hinges on the specific scenario.”

Fig. 2. Breakdown of the 10-year total cost of ownership. Results are presented for an average (A) passenger car, (B) light duty truck, (C) medium duty vehicle, and (D) heavy duty vehicle in the contiguous United States. The vehicle types include electric vehicles charged via Direct Current Fast Charging (DCFC-EV), Battery Swapping (BSS-EV), and Dynamic Wireless Power Transfer (DWPT-EV). The EVs are compared to an average internal combustion engine vehicle (ICEV) and hybrid electric vehicle (HEV) from each vehicle category. The baseline scenarios are shown in this static figure and all scenarios are shown in the interactive figure (see Supplementary Note 1) or repository.

Supplementary Figure 10. Breakdown of the 10-year total cost of ownership in 2022. Results are presented for an average (A) passenger car, (B) light duty truck, (C) medium duty vehicle, and (D) heavy duty vehicle in the contiguous United States. The vehicle types include an average electric vehicle (EV), internal combustion engine vehicle (ICEV), and hybrid electric vehicle (HEV) from each vehicle category.

Supplementary Figure 11. Breakdown of the total vehicle greenhouse gas emissions in 2022 and 2050. The emissions are presented for optimistic (Opt.), baseline (Base.), and conservative (Cons.) electric vehicle adoption (Adopt) and electricity mix (Mix) scenarios. Vehicle emissions from the contiguous United States (U.S.) are shown for electric vehicles (EVs) charged via Direct Current Fast Charging (DCFC), Battery Swapping (BSS), and Dynamic Wireless Power Transfer (DWPT). Results are compared to an average internal combustion engine vehicle (ICEV) and hybrid electric vehicle (HEV). The results are broken down for passenger cars (Car), light-duty trucks (LDT), medium-duty vehicles (MDV), and heavy-duty vehicles (HDV).

REVIEWERS' COMMENTS

Reviewer #1 (Remarks to the Author):

The authors have made substantial updates to the manuscript and addressed each comment thoroughly. Thank you. Recommending move to publication.

Reviewer #3 (Remarks to the Author):

The authors answered all the comments of reviewers, hence the paper can be accepted. Now, the paper quality has been increased.

REVIEWERS' COMMENTS

Reviewer #1 (Remarks to the Author):

The authors have made substantial updates to the manuscript and addressed each comment thoroughly. Thank you. Recommending move to publication.

Reviewer #3 (Remarks to the Author):

The authors answered all the comments of reviewers, hence the paper can be accepted. Now, the paper quality has been increased.

We thank the reviewers for their insightful comments throughout the peer-review process.